

# Effects of perceptual similarity but not semantic association on false recognition in aging

Kayleigh Burnside[1,*], Caroline Hope[2,*], Emma Gill[2] and Alexa M. Morcom[1,3]

[1] Psychology, University of Edinburgh, Edinburgh, United Kingdom
[2] Edinburgh Medical School, University of Edinburgh, Edinburgh, United Kingdom
[3] Centre for Cognitive Ageing and Cognitive Epidemiology, University of Edinburgh, Edinburgh, United Kingdom
[*] These authors contributed equally to this work.

## ABSTRACT

This study investigated semantic and perceptual influences on false recognition in older and young adults in a variant on the Deese-Roediger-McDermott paradigm. In two experiments, participants encoded intermixed sets of semantically associated words, and sets of unrelated words. Each set was presented in a shared distinctive font. Older adults were no more likely to falsely recognize semantically associated lure words compared to unrelated lures also presented in studied fonts. However, they showed an increase in false recognition of lures which were related to studied items only by a shared font. This increased false recognition was associated with recollective experience. The data show that older adults do not always rely more on prior knowledge in episodic memory tasks. They converge with other findings suggesting that older adults may also be more prone to perceptually-driven errors.

## INTRODUCTION

It is well established that memory for events is impaired in even healthy aging (*Light, 1991*). An emerging hypothesis is that prior knowledge, which is well maintained in older age, can support declining episodic memory to some degree (*Naveh-Benjamin et al., 2003*; *Castel, 2005*; *Umanath & Marsh, 2014*). However, this support may also be a "double-edged sword" (*Reder et al., 2007*), bringing a greater cost in memory errors than in the young. Consistent this view, older adults appear to be particularly susceptible to semantic as opposed to perceptual influences on false memory for categorically related similar pictures (*Koutstaal et al., 2003*; *Pidgeon & Morcom, 2014*). To investigate whether the same pattern would be found in a verbal false memory task, we contrasted the effects of associative and perceptual relatedness on false recognition in older and young adults in two experiments.

Although older adults are widely regarded as more prone to false memory (e.g., *Devitt & Schacter, 2016*; *Mitchell & Johnson, 2009*; *Schacter, Koustaal & Norman, 1997*), the reality may be more nuanced. In the categorized pictures paradigm, older adults are more likely

Corresponding author
Alexa M. Morcom,
alexa.morcom@ed.ac.uk

to falsely recognize pictures of objects they have not studied when the lures are related to studied items by membership of the same basic-level category (e.g., a picture of a different dog; *Koutstaal & Schacter, 1997*; *Koutstaal et al., 1999b*; *Lovden, 2003*). These errors are attributed to a greater reliance on processing semantic gist compared to specific item and contextual information (*Koutstaal & Schacter, 1997*). Older adults are also often reported to show increased false recognition when lures are associatively rather than categorically related to studied items, in the Deese-Roediger-McDermott paradigm (*Deese, 1959*; *Roediger & McDermott, 1995*; *Intons-Peterson et al., 1999*; *Norman & Schacter, 1997*; *Schacter, Israel & Racine, 1999*; see also *Rankin & Kausler, 1979*), although more consistently so when false memory is tested by recall than by recognition (see *Gallo, 2006*, pp. 184–202 for review). This, too, has been attributed to an increase in gist-based memory (*Tun et al., 1998*; *Brainerd & Reyna, 2002*).

There have been few direct investigations of the specific roles that semantic and perceptual information play in these memory errors, as lures often share both types of relations with studied items. *Koutstaal et al. (2003)* found heightened false recognition in older adults for pictures of unstudied objects sharing basic-level category membership with studied items (e.g., *candles*), but not for items from visually similar 'abstract' novel object sets. A similar effect was observed when novel objects were given verbal labels, leading the authors to propose that older adults rely more on overt verbal semantic categorization than the young. We recently replicated *Koutstaal et al.*'s (*2003*, Experiment 2) finding of a stronger age-related increase in false recognition for pre-experimentally meaningful picture lures compared to lures which were members of novel, 'abstract' categories, consistent with specific importance of semantic relatedness. However, unlike Koutstaal et al., we also found an age-related increase in errors for the 'abstract', visually similar lures (*Pidgeon & Morcom, 2014*). It is possible that the latter was due to greater incidental semantic processing of the multi-item 'abstract' categories by older adults. However, converging evidence that their greater susceptibility to false recognition is not driven entirely by semantic processing comes from studies finding age-related increases in false recognition of phonologically similar as well as semantically associated lures (*Budson et al., 2003*; *Sommers & Huff, 2003*; *Watson, Balota & Sergent-Marshall, 2001*; see also *Rankin & Kausler, 1979*). To our knowledge, no study has specifically examined visual perceptual influences on false recognition in older adults. Thus, the nature of perceptual influences on false recognition in aging is currently unclear, as are the conditions under which older adults rely more on semantic information than the young.

The current study examined older and young adults' susceptibility to semantic and perceptual influences on false recognition using a variant on the DRM paradigm (Fig. 1). The manipulations of associative and perceptual relatedness using semantically associated words in fancy fonts were taken from *Arndt & Reder*'s (*2003*) study in young adults. Experiment 1's design was otherwise based on *Koutstaal et al.*'s study (*2003*; Experiment 2) and our subsequent replication and extension (*Pidgeon & Morcom, 2014*). This meant that study and test lists were intermixed rather than blocked, as is usual in the categorized pictures paradigm, unlike in the DRM paradigm, in which blocked (list-wise) presentation is typical. In the Correlated Font conditions, sets of associated DRM words were studied in the

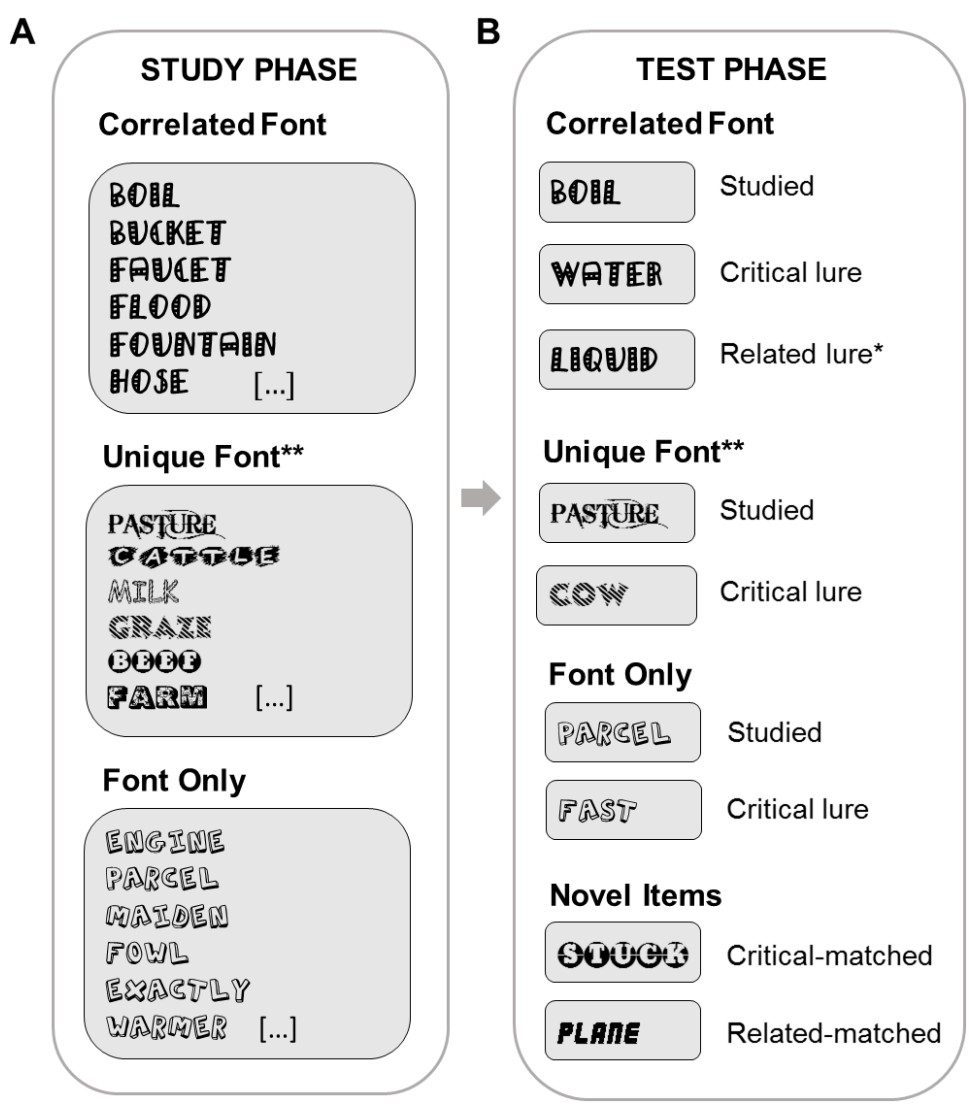

**Figure 1** **Design figure (new).** Design of Experiments 1 and 2. (A) partial example sets of studied words for each condition (see 'Materials'). (B) example studied and lure test words corresponding to these studied sets for each condition, and novel items. * indicates a condition used only in Experiment 1. ** indicates a condition used only in Experiment 2.

same font. At test, studied items and critical and related lures were also presented in this font. In the Font Only conditions, sets of unrelated items were studied in the same font. At test, studied items and lures were also presented in this font. Experiment 2 included an additional condition aimed at assessing interactions between font and semantic information.

Based on our earlier findings in the categorized pictures task, we expected that older adults would show an increase in false recognition for semantically-related than perceptually-related lures, relative to the young, and a (smaller) increase in false recognition of perceptually-related lures (*Koutstaal et al., 2003*; *Pidgeon & Morcom, 2014*). We were also interested in whether age-related differences in memory errors due to semantic
and perceptual relations were accompanied by a subjective experience of recollection. We measured the quality of recognition using the Remember-Know (RK) procedure (*Mandler, 1980*). This helps to disambiguate false memory from age-related differences in responses based directly on semantic knowledge or guessing. Previous work suggests that age-related increases in false recognition are partly driven by an increase in illusory recollection (*McCabe et al., 2009*). However, findings from DRM tasks have been mixed: *Intons-Peterson et al. (1999)* showed increased false recollection in older adults using the RK procedure, as did *Gallo & Roediger (2003)* using a source memory task, but two other studies using RK did not (*Schacter, Israel & Racine, 1999*; *Skinner & Fernandes, 2009*).

## EXPERIMENT 1

### Method

#### Participants

Participants were 24 young (19–23 years, $M = 20.4$) and 24 older adults (61–73 years, $M = 66.5$). We previously estimated that group sizes of 24 were necessary to replicate *Koutstaal et al.*'s (*2003*) critical Stimulus Type by Age interaction with .95 power (effect size $f = .513$; see *Pidgeon & Morcom, 2014*). This N would also yield power = .78 to detect an age-related increase in false recollection due to perceptual similarity (effect size $d = .704$ from *Pidgeon & Morcom, 2014*; one-tailed test). All participants were native English speakers with normal or corrected-to-normal vision. Baseline cognitive tests showed higher crystallized IQ in the older group than in the young, as expected (Wechsler Test of Adult Reading standardized score in young $M = 116$, older $M = 123$, SD = 6.5 and 5.0; $t(42.2) = 4.15$, $p < .001$, $d = 1.2$; *Holdnack, 2001*) and no difference in forward Digit Span ($M = 7.3$ and 7.5, SD = 1.1 and 1.3; $t(45.6) = .46$, $p = .650$, with moderate Bayesian evidence for the null hypothesis, $BF_{10} = 3.3$, range = 2.6 for prior width .5 to 4.0 for width .9). The older group showed a typical decrement on Digit Symbol Coding (Wechsler Adult Intelligence Scale IV; *Wechsler, 2008*; young $M = 71.3$, old = 56.9, SD = 12.3 and 10.3; $t(44.7) = 4.42$, $p < .001$, $d = 1.3$) and were slower on part B relative to part A of the Trail Making test (*Reitan, 1958*; for difference, $t(32.8) = 4.0$, $p < .001$, $d = 1.2$; consistent with reduced executive function and processing speed (*Salthouse, Atkinson & Berish, 2003*). Written consent was obtained from all participants. The study was approved by the Psychology Research Ethics Committee, University of Edinburgh (ref. 72-1314/2).

#### Materials

Stimuli were constructed using words written in distinctive fonts. The three experimental conditions are illustrated in Fig. 1. In the Correlated Font conditions, sets of associated words from DRM lists were studied in the same font, and lures were related to studied items by both semantic association and font. Two types of lure were included at test, making two Correlated Font conditions: those with DRM critical associates as lures (Correlated Font-Critical), and those with unstudied associates from the studied list as lures (Correlated Font-Related). In the Font Only condition, sets of unrelated words were studied in the same font, and additional unrelated words served as lures and were presented in the same font at test.

We used 512 words and 80 distinctive fonts (from http://www.urbanfonts.com and http://www.1001freefonts.com/). Thirty two 11-word associated DRM lists were drawn from the University of South Florida (USF) Free Association Norms database http://w3.usf.edu/FreeAssociation/; (*Nelson, McEvoy & Schreiber, 2004*), selected to maximize Backward Associative Strength (BAS) from list items to their Critical lures (mean = .48). Mean BAS for list words to other list words used as Related lures was .0065. Sets of 11 unrelated words were also constructed using words from the MRC Psycholinguistic database (*Wilson, 1988*), selected to match the associated DRM words on word length and frequency (median length = 6 and written frequency = 32). Each study phase list comprised 432 items, including nine words from each of 16 associated lists per DRM condition (16 lists for Correlated Font-Critical and 16 for Correlated Font-Related), and nine words from each of 16 unrelated sets of words for the Font Only condition. Each test phase list comprised 128 items, and included one studied word and one lure for each studied list (16 studied items and 16 lures per condition). Test lists also contained 32 Novel words. Sixteen of the Novel words were separately taken from the MRC database to match the Critical lures on word length and frequency (Critical-matched; median length = 5 and written frequency = 110 per million). The other 16 Novel words were taken from the unrelated word sets, and therefore matched to the studied associated items, related lures in the Correlated Font-Related condition, and unrelated words in the Font Only condition. To counterbalance the stimuli over participants, we randomly allocated fonts and words to conditions several times, and randomly allocated the counterbalanced lists to participant numbers. First, DRM list words were allocated to serve as studied items for the Critical and Related lure conditions, and Related lures. Next, unrelated words were allocated to serve as studied (Font Only) or Novel items. Fonts were then allocated to the associated and unrelated sets of words, and to studied and Novel items. Young and older groups received precisely the same lists.

### Procedure

Participants completed a single study-test cycle after a short practice. They viewed single words in distinctive fonts at the center of a computer screen. At study, they completed an incidental encoding task, judging the degree to which the word's font fit its meaning with a 4-way button press response (Very Well to Very Poorly). At test, participants were asked to "decide which words are new and which are old and what, if anything you remember about the words you have seen before", with no instruction regarding font. We used a variant of the Remember/Know procedure with standard instructions (*Gardiner, 1988*; *Tulving, 1985*), though we replaced the term 'Know' with 'Familiar' for clarity (see *Migo, Mayes & Montaldi, 2012*). Participants judged whether they recollected seeing a word ('Remember'; R), or whether it was just familiar ('Familiar'; F), or new ('New'), or they could not decide whether it was studied ('Guess'). At study and test, items remained on-screen until a response was made for up to 7,000 msec, followed by 1,500 msec fixation.

**Table 1 True and False Recognition as a function of age, condition, and item type in Experiment 1.**
(Means, with SD in parentheses). Overall Recognition proportions are the proportions of items in each condition which were judged old (both "Remember" and "Familiar" responses). Adjusted recognition scores are the raw Overall scores after subtraction of the raw scores for the corresponding Novel item condition (see 'Method' for details of conditions and measures).

|  | Overall recognition | | Adjusted recognition | |
|---|---|---|---|---|
|  | **Young** | **Older** | **Young** | **Older** |
| **Correlated Font** | | | | |
| Studied item true recognition: | .79 (.14) | .74 (.17) | .75 (.16) | .66 (.30) |
| List lure false recognition: | .34 (.17) | .45 (.29) | .29 (.18) | .30 (.23) |
| Critical lure false recognition: | .57 (.19) | .52 (.28) | .53 (.18) | .36 (.24) |
| **Font Only** | | | | |
| Studied item true recognition: | .70 (.17) | .65 (.24) | .59 (.14) | .50 (.25) |
| Lure false recognition: | .14 (.11) | .30 (.29) | .09 (.10) | .15 (.18) |
| **Novel** | | | | |
| List-Matched false recognition: | .04 (.06) | .15 (.28) | | |
| Critical-Matched false recognition: | .04 (.08) | .16 (.26) | | |

## Data analysis

All degrees of freedom and $p$ values were Greenhouse-Geisser corrected, and Welch's unequal variances $t$-test was used where appropriate. We also supplemented null-hypothesis significance testing with Bayes Factor (BF) analyses using JASP (https://jasp-stats.org/; version 0.8.0.1; *Rouder et al., 2009*). Bayes Factors were used to provide evidence for the null hypothesis where this was important for interpretation of findings, and for summary analyses across experiments. Although categorization of levels of Bayesian evidence is necessarily arbitrary, for clarity we adopt the labels used in JASP (*Lee & Wagenmakers, 2013*), i.e., that BF <3 indicates "anecdotal" evidence, 3 <BF <10 "moderate", 10 <BF <30 "strong", 30 <BF <100 "very strong", and BF >100 "extreme" evidence). The Bayesian $t$-tests used uninformative Cauchly priors with $M = 0$, width = .71, with additional robustness checks of BFs under a range of prior distribution widths (width = .5 or .9) to check that this did not substantially change the results (a wider prior increases evidence for the null hypothesis).

## Results

Table 1 summarizes the overall response proportions (collapsed over R and F), and Figs. 2A–2B illustrates true and false recollection. To assess true and false recognition we analysed both raw response proportions and proportions of hits (correctly identified studied items) and false alarms (FA) after adjusting for FA to novel items (i.e., subtracting novel FA proportions; see e.g., *Koutstaal et al., 2003*). For hits, the corrected measure was equivalent to the $P_r$ discrimination index which assumes two-high-threshold dual process recognition (*Snodgrass & Corwin, 1988*). Hit proportions for Correlated Font items were collapsed over the Critical and List conditions (as the studied items were equivalent). For Critical lures, the corresponding Novel items were Critical-Matched, and for studied items and for List and Font Only lures they were Related-matched (see 'Materials'). Lastly, we

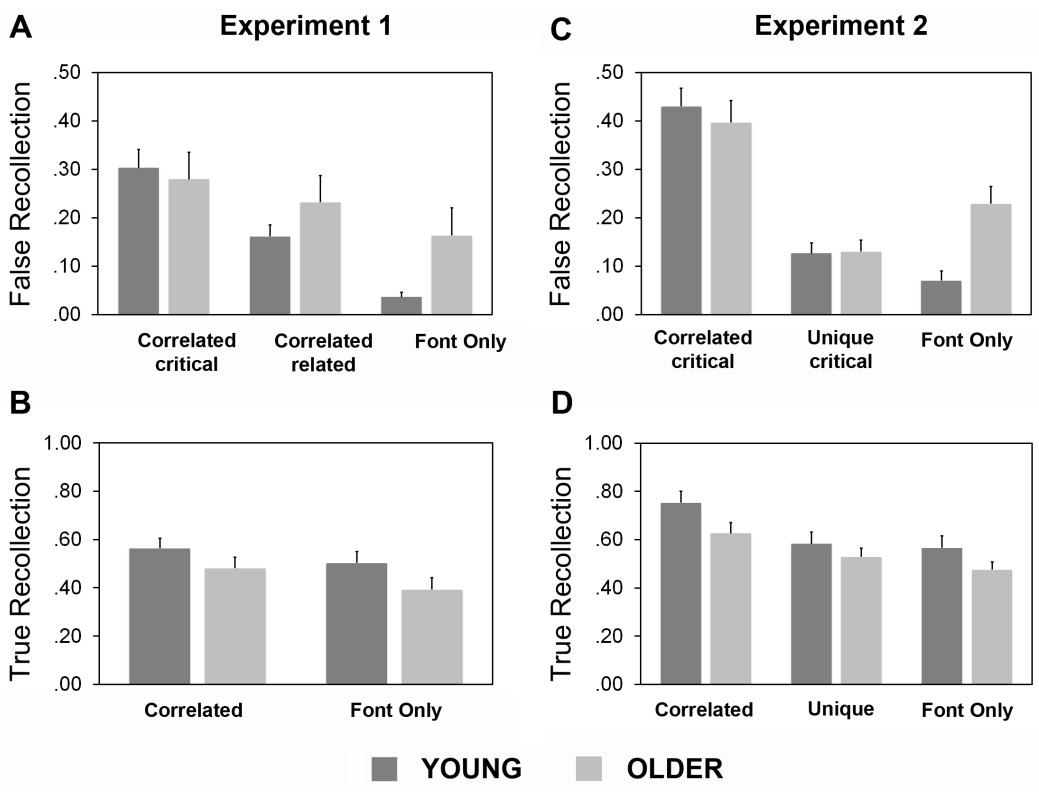

**Figure 2  True and false recollection.** (A) (False Recollection) and (B) (True Recollection) show data from Experiment 1, and (C) (False Recollection) and (D) (True Recollection) show data from Experiment 2. True Recollection is the proportion of studied items which were (correctly) recollected. False Recollection is the proportion of unstudied lures which were (incorrectly) recollected. Bars show the mean proportions of recollected items, and error bars represent SE.

assessed recollection, using the proportion of hits and lure false alarms which attracted "Remember" (R) responses (Figs. 2A and 2B).

### True recognition

True recognition was better in both age groups for items in the Correlated Font than the Font Only condition, and better on one of the measures in the young than the older group. ANOVA on raw hits with factors of Condition (Correlated Font/ Font Only) and Group (Young/Older) showed a main effect of Condition, $F(1, 46) = 17.10$, MSE $= .18$, $p < .001$; $\eta_p^2 = .27$; see Table 1 and Fig. 2. For raw proportions of hits to studied items it was unclear whether or not the groups differed: for main effect of Group, $F(1, 46) = .99$, MSE $= .06$, $p = .325$, $BF_{01}$ for the null hypothesis $= 1.9$; for Condition $\times$ Group, $F(1, 46) = .02$, MSE $< .001$, $p = .89$. Discrimination of studied from new items ($P_r$) was also greater in the Correlated Font condition, $F(1, 46) = 18.23$, $p < .001$, $\eta_p^2 = .28$, MSE $= .19$, and unlike for raw recognition this adjusted measure showed greater true recognition in the young group, $F(1, 46) = 6.48$, MSE $= .58$, $p = .014$, $\eta_p^2 = .12$. The interaction was not significant, $F(1, 46) = .02$, MSE $< .001$, $p = .90$, although there was no Bayesian evidence against one, $BF_{01} = .95$. For true recollection, the Correlated Font and Font Only conditions again

differed (for main effect, $F(1, 46) = 23.4$, MSE $= .27$, $p < .001$, $\eta_p^2 = .34$, but age effects for this measure were unclear: for main effect, $F(1, 46) = 1.22$, MSE $= .01$, $p = .28$, with anecdotal Bayesian evidence for the null, $BF_{01} = 1.3$; for Condition $\times$ Group, $F(1, 46) = 1.7$, MSE $= .17$, $p = .19$. The presence of only minor age differences in true recognition meant that effects on false recognition were more straightforward to interpret.

### False recognition

For Novel items, ANOVA on proportions falsely recognized with factors of Condition (Critical-Matched/ Related-matched) and Group (Young/Older) showed a slightly higher baseline false alarm rate in the older group, reflected in a main effect of Group, $F(1, 46) = 4.20$, $p = .047$; $\eta_p^2 = .08$, MSE $= .32$. The conditions did not differ significantly, for main effect of Condition, $F(1, 46) < .001$, $p = 1.00$, nor did the interaction, $F(1, 46) = .60$, $p = .442$, MSE $= .002$, although Bayesian evidence against the latter effect was only anecdotal, with $BF_{01}$ for the null hypothesis $= 2.3$. No condition or group effects were significant for proportions of falsely recollected novel items either, and there was strong Bayesian evidence against an interaction (for main effect, $F(1, 46) = 2.43$, MSE $= .002$, $p = .126$, BF against inclusion in model $= 3.7$; for interaction, $F(1, 46) = .09$, MSE $= .00007$, $p = .768$, BF against inclusion $= 55.6$). The presence of group differences in raw Novel FA suggests caution may be needed in interpretation of Novel-adjusted false recognition measures, even though this effect did not differ by condition.

Across the two groups, lure false recognition was highest for Correlated Font Critical lures, intermediate for Correlated Font Related lures and lowest for Font Only lures (Table 1 and Fig. 2). The effects of age Group also depended on Condition for both raw and adjusted false recognition. ANOVA on raw lure FA with factors of Condition (Critical lure/ Related lure/ Font Only lure) and Group (Young/ Older) revealed a main effect of Condition, $F(1.7, 77.6) = 63.1$, MSE $= .23$, $p < .001$, $\eta_p^2 = .53$, and interaction with Group, $F(1.7, 77.6) = 9.97$, $p < .001$, $\eta_p^2 = .18$, MSE $= .23$; for main effect of Group, $F(1, 46) = 1.5$, MSE $= .19$, $p = .23$. *Post hoc* tests showed a significant increase in Font Only lure false recognition in the older group compared to the young, $t(30.6) = -2.88$, $p = .007$, $d = .83$; alpha $=. 017$. In the Correlated Font conditions, age effects on false recognition were non-significant for both Critical and Related lures, although Bayes Factors showed only anecdotal evidence for the null hypothesis, $t(41.2) = .89$, $p = .38$, $BF_{01} = 2.5$; $t(36.8) = -1.65$, $p = .108$; $BF_{01} = 1.2$. These findings suggested that older adults showed an increase only in perceptually-driven false recognition, but we also analyzed false recollection and adjusted measures to rule out effects of response criterion.

For false recollection ANOVA with the same factors again showed an interaction of Condition $\times$ Group, $F(1.7, 80.4) = 6.0$, MSE $= .088$, $p = .005$, $\eta_p^2 = .12$, as well as a main effect of Condition, $F(1.7, 80.4) = 30.0$, MSE $= .44$, $p < .001$, $\eta_p^2 = .37$, but not Group, $F(1, 46) = 1.0$, MSE $= .12$, $p = .32$. *Post hoc* tests did not show a significant age-related increase in any condition (for Font Only lures, $t(24.3) = -2.31$, $p = .030$; for Critical lures, $t(40.6) = .40$, $p = .695$, for Related lures, $t(31.3) = 1.13$, $p = .268$; alpha $= .017$). Bayes factors showed anecdotal evidence for the alternative hypothesis in the Font Only condition, $BF_{10} = 2.4$, (range $= 2.5$ for prior width .5 to 2.2 for width .9), and moderate Bayesian

evidence for the null hypothesis for the Critical lures, $t(40.1) = .40$, $p = .70$, $BF_{01} = 3.3$ (range $= 2.5$ to $4.0$), but anecdotal evidence for the Related lures, $t(31.3) = -1.12$, $p = .27$, $BF_{01} = 2.1$. Although this pattern qualitatively supported the results for raw FA, there was not enough evidence to clearly determine whether the increase in false recognition of Font Only lures by older compared to young adults was accompanied by an increase in false recollection.

The comparison between conditions was complicated by the findings for adjusted false recognition (Table 1). ANOVA again showed a main effect of Condition, $F(1.8, 82.2) = 61.35$, $p < .001$, $\eta_p^2 = .57$, $MSE = 1.43$, and an interaction of Condition $\times$ Group, $F(1.8, 82.2) = 12.00$, $p < .001$, $\eta_p^2 = .21$, $MSE = .28$; for main effect of Group, $F(1, 46) = 1.2$, $MSE = .04$, $p = .28$, but *post hoc* tests showed that this reflected group differences for Critical rather than Font Only lures. While older adults falsely recognized a similar (though numerically larger) number of Font Only Lures to the young, $t(46) = 1.69$, $p = .097$, $BF_{01}$ for the null hypothesis $= 1.0$, on this measure they falsely recognized significantly *fewer* Critical Lures, $t(46) = 3.03$, $d = .78$, $p = .004$, $BF_{10}$ for the alternative hypothesis $= 6.8$, range $= 6.7$ for prior width $= .5$ to $6.6$ for width $= .9$; for Related lures, $t(46) = .07$, $p = .944$, $BF_{01} = 3.4$, range $= 2.7$ to $4.2$; alpha $= .017$. This apparent age-related reduction in adjusted false recognition had not been predicted.

## Discussion

The results show an age-related increase in perceptual but not associative false recognition in this task. Counter to our prediction, age effects on false recognition and recollection were less prominent for the semantically and perceptually related Correlated Font Critical lures than for the perceptually-related Font Only lures. However, because of the different finding for the corrected FA measure, it is not possible to determine unambiguously which condition carried the age effect.

While numerous previous studies have reported age-related increases in associative false recognition, others have also found null effects (e.g., *Kensinger & Schacter, 1999*; *Gallo & Roediger, 2003*; *Kensinger & Schacter, 1999*; *Intons-Peterson et al., 1999*; *Skinner & Fernandes, 2009*; see *Gallo, 2006*, pp. 193–189). Some of the differences between studies, and the present discrepancy between raw and adjusted false recognition findings, probably reflect differences in the selection of items for baseline (novel) conditions. It is important to demonstrate that age-related differences in false recognition reflect differences in memory rather than in decision criterion (see *Roediger et al., 2001*). One way to do this is to adjust for baseline FA to novel items (e.g., *Koutstaal et al., 2003*). It is common for DRM studies to counterbalance stimuli so that list items and critical lures from unstudied lists serve as Novel items. However, the Novel items are typically either DRM list items from different list positions from those studied, or a mix of unstudied list items and unstudied critical lures (examples of exceptions: *McCabe & Smith, 2002*; *Schacter, Israel & Racine, 1999*). Therefore, adjusted critical lure false recognition may often not fully correct for baseline age-related differences in responding, because of differences between critical lures and novel items on indices such as familiarity and imageability. In the present study we did not counterbalance Novel items and lures but did match them on word length and frequency.

Thus, the apparent age-related reduction in adjusted Critical lure false alarms was likely influenced by differences in young and older adults' responses due to unmeasured item characteristics between the Critical Lure and Critical-Matched Novel conditions. More importantly, it was possible that the finding of a differential age effect on associative and perceptually-driven false recognition reflected subtle differences in the two groups' false recognition of Critical-Matched and Related-Matched Novel items. There was no clear evidence for this, as age effects on Novel false alarms were equivalent in the two conditions. However, replication in a study with fully counterbalanced and matched Novel items was essential.

We also wanted to test a possible mechanism of an age-related reduction in associative false recognition in this task, if present. Two known modifiers of age-related differences in false memory are individual differences in prefrontal cortex function and the post-retrieval monitoring demands of the task (*Gallo, 2006*). These are unlikely to account for the current data. The older group had high verbal IQ, but still showed typical reductions on measures of processing speed and executive function. Thus, they were unlikely to be better than the young at retrieval monitoring. However, reduced activation at encoding was a possibility. Lure activation is thought to be automatic and preserved in old age (*Gallo & Roediger, 2003*; *Koutstaal, 2003*), but can also be influenced by processing at encoding. In young adults, false recognition is lower when list items have unique as opposed to shared study context, in terms of either list blocking (*Goodwin, Meissner & Ericsson, 2001*) or font (*Arndt & Reder, 2003*). Such contextual effects may be greater in older adults, as reflected in greater impact of blocking or task manipulations (*Taconnat et al., 2006*; *Thomas & Sommers, 2005*; *Tun et al., 1998*; see also *Thomas, Bonura & Taylor, 2012*). In the present study, although associated lists were presented in the same fonts, it was possible that the judgement of specific "fit" of each unique item to the font emphasized older adults' item-specific processing at the expense of relational processing. If so, studying each word in a unique font should exaggerate this "reversed" age-related difference.

## EXPERIMENT 2

In this experiment, we contrasted false recognition of perceptually-related and semantically-associated lures in a fully counterbalanced as well as matched design. *Arndt & Reder (2003)* previously showed that adjusted critical lure false recognition was greater when list items were studied in the same font (Correlated Font condition) than when they were each studied in a different font (Unique Font; see Fig. 1). We expected that if the age-related reduction in critical lure false recognition in Experiment 1 was due to greater item-specific encoding leading to reduced lure activation in the older adults, it would be more pronounced in the Unique Font relative to the Correlated Font conditions. If not, the logic used prior to Experiment 1 predicted an increase in Correlated Font false recognition in the older group. We also predicted that adjusted Font Only lure false recognition would again be greater in the older adults. In both cases, we also expected age-related increases in false recollection.

## Method

### *Participants*

Participants were 24 young (18–33 years, $M = 22$; 17 female) and 24 older (60–75 years, $M = 67$; 18 female) adults. All were native English speakers with normal or corrected-to-normal vision. As in Experiment 1, and as expected, the older group had higher crystallized IQ than the young (WTAR standard score: $M = 110$ and 124, SD = 7 and 3; $t(46) = 8.52$, $p < .001$, $d = 2.5$; 1 missing value), and greater verbal fluency on the FAS Controlled Word Association Test (*Lezak et al., 2012*); $t(36.8) = 6.52$, $p < .001$, $d = 1.9$, $M = 47$ and 62 words, SD = 6 and 10). Despite this, and as in Experiment 1, they showed a typical age-related decrement in processing speed (Digit Symbol Coding completion time young $M = 202$ s, older $= 239$ s, SD $= 21$ and 35; $t(38.0) = 4.55$, $p < .001$, $d = 1.3$) and were slower on part B relative to part A of the Trail Making test (for difference $t(33.7) = 2.49$, $p = .018$, $d = .72$; $M = 22$ and 31 s, SD $= 8$ and 16 s). The groups did not differ significantly in Digit Span $(43.8) = 1.98$, $p = .054$, although Bayesian evidence did not provide support for the null hypothesis either, $BF_{01} = .72$). Written consent was obtained from all participants. The study was approved by the Psychology Research Ethics Committee, University of Edinburgh (ref. 80-1516/2).

### *Materials*

Words were selected from 168 9-item DRM associate lists from the USF Free Association Norms database (*Nelson, McEvoy & Schreiber, 2004*), and 157 distinctive fonts, also from the same sources as Experiment 1. The lists were selected to maximize BAS to their critical lures ($M = 0.32$). In two conditions (see Fig. 1 for design), sets of semantically associated items were studied, and only Critical lures included at test (Correlated Font and Unique Font). In the third, Font Only condition, sets of semantically unrelated items were again studied in the same font, with unrelated lures in the same font at test, as in Experiment 1. In this fully counterbalanced and matched design, we again used two types of Novel items to calculate the baseline false alarm rates. For the Correlated Font and Unique Font conditions, in which Critical lures were shown, the corresponding Novel items were critical lures from unstudied DRM lists (Critical-matched). For the Font Only condition, in which lure words were (non-critical) items from unstudied DRM lists, the Novel items were drawn from the same sets of unrelated items as the studied items and lures (Related-matched).

Each study phase comprised 324 items, including nine words from 12 associated DRM word lists for the Correlated Font and Unique Font conditions, and 1 word from each of a further 84 associated word lists, grouped into sets of 9 for the Font Only condition (inspected to check they were not obviously related). In the Correlated Font and Font Only conditions, all words in a 9-item list (related or not) were presented in the same font (12 per condition). In the Unique Font condition, each word was presented in a unique font (84 further fonts) (total = 324 items). Each test list comprised 98 items, including 36 studied words (one from each studied set in each of the three conditions), 24 Critical lures (12 each for the Correlated Font and Unique Font conditions), and 12 unrelated Font Only lures. Studied items were presented in the studied font in all conditions. The Correlated Font Critical lures and the Font Only lures were also presented in the same font

**Table 2** **True and False Recognition as a function of age, condition, and item type in Experiment 2.** (Means, with SD in parentheses). Overall Recognition proportions are the proportions of items in each condition which were judged old (both "Remember" or "Familiar" responses). Adjusted recognition scores are the raw Overall scores after subtraction of the raw scores for the corresponding Novel item condition (see 'Method' for details of conditions and measures).

| | Overall recognition | | Adjusted recognition | |
|---|---|---|---|---|
| | Young | Older | Young | Older |
| **Correlated Font** | | | | |
| Studied hits: | .91 (.14) | .88 (.11) | .87 (.14) | .78 (.18) |
| Critical lure false alarms: | .71 (.15) | .71 (.23) | .61 (.20) | .61 (.23) |
| **Unique Font** | | | | |
| Studied hits: | .78 (.18) | .78 (.15) | .75 (.21) | .68 (.20) |
| Critical lure false alarms: | .27 (.13) | .41 (.22) | .18 (.17) | .31 (.23) |
| **Font Only** | | | | |
| Studied hits: | .75 (.16) | .76 (.19) | .71 (.18) | .66 (.21) |
| Lure false alarms: | .25 (.15) | .48 (.23) | .22 (.17) | .43 (.17) |
| **Novel** | | | | |
| Unrelated-matched false alarms: | .03 (.06) | .10 (.17) | | |
| Critical-Matched false alarms: | .10 (.11) | .10 (.19) | | |

as their corresponding studied items. The Unique Font Critical lures were presented in the same font as one of their corresponding studied items. The 12 Critical-matched and 12 Related-matched Novel items were each presented in an unstudied font. Allocation of fonts and lists to conditions was fully counterbalanced.

### Procedure

The procedure was identical to that of Experiment 1.

### Data analysis

The data analysis followed the same procedures as Experiment 1.

## Results

Table 2 lists the raw and adjusted response proportions (collapsed over R and F), and Figs. 2C–2D illustrates the true and false recollection results.

### True recognition

Across both age groups participants were more likely to correctly recognize items from semantically associated lists which had been studied in the same font (Correlated Font) than those studied in unique fonts (Unique Font), or unrelated items which had been studied in the same font (Font Only) (Table 1 and Fig. 2). For hits, ANOVA with factors of Condition and Group revealed only a main effect of condition, $F(2.0, 90.8) = 15.2$, MSE $= .27$, $p < .001$, $\eta_p^2 = .25$; for Group main effect, $F(1, 46) = .03$, MSE $= .001$, $p = .87$, BF against inclusion in the model $= 4.1$; for interaction, $F(2.0, 90.8) = .25$, MSE $= .004$, $p = .78$, BF $= 9.1$ against inclusion in the model. *Post hoc* tests confirmed differences between Correlated Font and both the Unique Font and the Font Only conditions, $t(47) = 4.47$, $p < .001$, $d = .65$ and $t(47) = -5.12$, $p < .001$, $d = .74$, but not between the Unique Font and Font

Only conditions, $t(47) = 1.02$, $p = .314$; alpha $= .017$, $BF_{01}$ for null hypothesis $= 3.9$ (range $= 2.9$ for prior width $= .5$ to $4.9$ for width $.9$). Analysis of adjusted true recognition (using false alarms to the "unrelated" Novel items drawn from unstudied DRM lists) showed the same pattern with a significant main effect of Condition only, $F(2.0, 90.1) = 15.3$, MSE $= .271$, $p < .001$, $\eta_p^2 = .25$, and non-significant Group main effect, $F(1, 46) = 2.54$, MSE $= .17$, $p = .118$, although only anecdotal Bayesian evidence for the null, $BF_{01} = 1.3$. The pattern was the same for true recollection (for main effect of Condition, $F(1.9, 85.4) = 15.7$, MSE $= .41$, $p < .001$, $\eta_p^2 = .25$; for Group, $F(1, 46) = 3.3$, MSE $= .29$, $p = .078$, BF for inclusion in the model $= 1.1$; for interaction, $F(11.9, 85.6) = .67$, MSE $= .02$, BF against inclusion in the model $= 2.5$). Therefore, as in Experiment 1, older adults' true recognition was similar to that of the young. Participants in both groups were more likely to recognize studied items which were semantically associated with others on the list than those only printed in the same font. We also did not find evidence for (or clearly against) an age-related boost to true memory from the semantic relations.

### False recognition

Baseline false recognition of novel items was higher for Critical-Matched than Related-matched items, with no difference between age groups (Table 2 and Fig. 2). ANOVA with factors of Condition and Group showed only a main effect of condition, $F(1, 46) = 4.7$, MSE $=. 031$, $p = .036$, $\eta_p^2 = .09$. Although the main effect of Group was not significant, $F(1, 46) = .862$, MSE $= .028$, $p = .358$; $F(1, 46) = 3.24$, MSE $= .021$, $p = .079$, Bayesian ANOVA did not show evidence against inclusion of this factor either, $BF_{01} = 1.1$. Proportions of novel items which were falsely recollected did not differ significantly according to Group or Condition. The possibility of subtle group differences in Novel FA might complicate interpretation of Novel-adjusted false recognition measures as in Experiment 1. However, unlike in Experiment 1, findings from the different measures converged, as outlined below.

Across the two groups, false recognition was higher in the Correlated Font condition than the two other conditions, but the groups also differed according to condition (Table 2 and Fig. 2). ANOVA with factors of Condition and Group showed a significant interaction, $F(1.9, 89.2) = 5.6$, MSE $= .15$, $p = .006$, $\eta_p^2 = .11$, as well as main effects of Condition, $F(1.9, 89.2) = 78.3$, MSE $= 2.1$, $p < .001$, $\eta_p^2 = .63$, and Group, $F(1, 46) = 9.3$, MSE $= .56$, $p = .004$, $\eta_p^2 = .17$). *Post hoc* tests showed significant age-related differences in the Font Only Condition, $t(40.0) = -3.99$, $p < .001$; $d = 1.15$ and the Unique Font condition, $t(37.3) = -2.6$, $p = .013$, $d = .75$; alpha $= .017$, but not in the Correlated Font condition, $t(39.7) = -.15$, $p = .88$, in which there was moderate evidence in favour of the null, $BF_{01} = 3.4$ (range $= 2.67$ for prior width $= .5$, $4.3$ for prior width $= .9$. This replicated the results of Experiment 1 for the two shared conditions, showing evidence against an age-related increase in associatively-driven false recognition because the groups did not differ for the Correlated Font lures. At the same time, there was again an increase in false recognition of Font Only lures in older adults. So far, findings were consistent with a greater tendency to perceptually-driven false recognition, but not semantically-driven false recognition, in older adults.

The false recollection results were similar to those for raw FA, except that the groups did not differ in the Unique Font condition. ANOVA showed an interaction of Condition $\times$ Group, $F(1.6, 74.6) = 8.78$, MSE $= .16$, $p < .001$, $\eta_p^2 = .16$, and *post hoc* tests showed group differences for Font Only, $t(36.9) = -3.91$, $p < .001$, $d = 1.13$, but not for Correlated Font, $t(44.5) = .56$, $p = .58$, $BF_{01} = 3.1$ (range $= 2.4$ for prior width $= .5$ to $3.7$ for width $.9$), or Unique Font, $t(45.2) = -.11$, $p = .92$; alpha $= .017$; $BF_{01} = 3.5$ (range $= 2.7$ for prior width $.5$ to $4.2$ for width $.9$). Again, these findings converged with those of Experiment 1 for the two shared conditions.

Unlike in Experiment 1, the findings for adjusted false recognition in the Correlated Font and Font Only conditions, after correcting for baseline false alarms to Novel items (Fig. 1), converged with those for overall false recognition and false recollection. ANOVA showed main effects of Condition, $F(2.0, 90.0) = 53.1$, $\eta_p^2 = .54$, MSE $= 1.90$, $p < .001$, and Group, $F(1, 46) = 7.07$, MSE $= .36$, $p = .011$, $\eta_p^2 = .13$. There was no clear evidence for an interaction, $F(2.0, 90.0) = 2.51$, MSE $= .09$, $p = .088$, BF for inclusion in the model $= 2.5$. Therefore, the older adults showed generally greater false recognition after correction for novel item false alarms, and the conditions did not differ. Age effects were also robust in the Font Only condition taken alone, consistent with Experiment 1's findings and our prediction of an age-related increase in perceptually-driven false recognition based on findings from the categorized pictures paradigm (*Pidgeon & Morcom, 2014*). Pairwise *post hoc* tests (alpha $= .017$) showed no group difference in the Correlated Font condition, $t(45.1) = .07$, $p = .95$; $BF_{01}$ in favour of null hypothesis $= 3.5$ (range $= 2.7$ with prior width $.5$ to $4.3$ with width $.9$), no clear evidence for or against a group difference in the Unique Font conditions, $t(41.3) = -2.28$, $p = .028$; $BF_{01} = .45$, but significantly higher false recognition in the older than the young adults in the Font Only condition, $t(45.6) = 4.41$, $p < .001$, $d = 1.3$.

### False recognition across experiments 1 and 2

To check the apparent convergence of the principal findings across the two experiments, we calculated across-experiment Bayes factors for comparisons of age-related differences in false recollection in the two shared conditions. We focus on the proportions of items falsely judged recollected, which were directly comparable across experiments and not biased by any differences in novel item processing. This measure is also indicative of effects on vivid false recollection rather than only on familiarity or guessing (the results were qualitatively similar for analyses of raw false recognition measures). As noted above, Bayes factors provide complementary information to $p$ values, indicating the strength of evidence for the null or the alternative hypothesis. A Bayes factor is also not affected by the number of statistical tests performed, unlike $p$ values (*Wagenmakers, 2007*). We found extreme evidence for greater false recollection of Font Only lures in the older relative to the young adults across experiments, $BF_{10}$ against the null hypothesis $= 299.6$ (range 278 for prior width $.5$ to 297 for width $.9$), 95% credible interval for false recollection proportions in young $= .03$ to $.08$ with $M = .053$; in older $= .13$ to $.26$ with $M = .198$. However, for Correlated critical lures there was evidence *for* the null hypothesis of no age-related difference in false recollection, $BF_{01}$ for null hypothesis $= 3.9$, range $= 2.9$ for prior width

.5 to 4.8 for width .9, 95% credible intervals for young $= .31$ to .41, for older $= .26$ to .41. When the two measures were entered into a Bayesian ANOVA with factors of Condition (Correlated/ Font Only), Group (Young/ Older) and Experiment (1/ 2), the Bayes factor in favor of inclusion of the interaction of Condition with Group was 2055.9. We did not find evidence for a difference between experiments in this effect, nor against one, with BF $=$ .84 against inclusion of the 3-way interaction. These results were qualitatively unchanged after exclusion of 3 older individuals whose overall false alarm rates were very high ($>.5$ for one or more Novel or Lure conditions; $BF_{10} = 187.9$ for an age-related increase in Font Only RFA, $BF_{10} = 445.8$ for Group $\times$ Condition, and $BF_{10} = 1.1$ for Group $\times$ Condition $\times$ Experiment). Together, the data therefore strongly supported our original hypothesis that older adults would show greater perceptually-driven false recollection (in the Font Only conditions). They also suggested that across experiments, false recollection due to associative semantic relations in this task was age-invariant.

## Discussion

The results of Experiment 2 converged with those of Experiment 1, again showing an age-related increase in lure false recognition. In this fully counterbalanced and matched design, in which item differences between novel and lure conditions could not contaminate age-related comparisons, we found converging results between the adjusted false recognition and the raw FA and false recollection measures. There was no hint of Experiment 1's 'reversed' age-related difference for the critical lures on the adjusted measure. Although age effects did not differ between conditions, it is notable that while the age-related increase in Font Only lure false recognition was clear cut, there was no evidence of an age effect on semantically-driven false recognition.

The results for the Unique Font condition were less clear than for the other two conditions, and are discussed further below. The main reason to include this condition was our prediction that if an age-related reduction in associative false recognition were again found in the Correlated Font condition in the second experiment, this reduction might reflect greater item-specific processing in older adults, expected to be enhanced further and therefore more pronounced for the Unique Font lures. Instead, there was no significant age effect in the Unique Font condition (nor clear evidence against one): an increase in false recognition in the older group was not robustly present after adjustment for baseline false alarms, nor present for false recollection responses.

## GENERAL DISCUSSION

In these experiments, older adults were more prone to perceptual false recognition than the young, and showed an increase in perceptual false recollection, but we did not find any increase in associative false recognition. The data are contrary to the general hypothesis that older adults falsely recognize more than the young because their episodic memory relies more on prior knowledge (*Koutstaal et al., 2003*; *Pidgeon & Morcom, 2014*; *Reder et al., 2007*). On this hypothesis, a greater age-related increase in false recognition was expected in the Correlated Font compared to the Font Only conditions. Instead, our results suggest that prior knowledge—in this case associations—is not always the main driver of

older adults' memory errors, and that perceptual information plays a role. We also found that these errors were accompanied by recollective experience, as predicted and in line with other evidence that older adults' memory errors are often accompanied by subjective recollection (see *Devitt & Schacter, 2016*; *McCabe et al., 2009*).

In the present study, older adults were about twice as likely as the young to falsely recognize unrelated lure words which were written in the same font as multiple studied words, and almost four times as likely to falsely recollect these lures. The data converge with our previous finding that older adults were more likely to falsely recognize lure pictures from pre-experimentally unfamiliar 'abstract' object categories (*Pidgeon & Morcom, 2014*). Both in that study and the present study, the age-related increase in false recognition was only observed for lures which resembled multiple studied items (see also *Koutstaal et al., 2003*; *Koutstaal et al., 1999a*; *Koutstaal et al., 1999b*; *Koutstaal & Schacter, 1997*). In our earlier study, one possibility is that this false recognition, and its increase in aging, was driven by the object-like nature of the 'abstract' images, some of which we found to be consistently nameable (*Pidgeon & Morcom, 2014*). Perceptual influences on false recognition are not well understood, but it is recognized that some form of prototype or gist extraction can proceed without prior knowledge, as shown by false recognition of abstract dot patterns which are similar to sets of studied patterns (*Posner & Keele, 1970*; *Gallo, 2006*, pp 35–36). However, apparent perceptual gist effects can also reflect *de novo* category learning that proceeds as multiple exemplars are encountered (*Stahl, Henze & Aust, 2017*, May 15). Potentially, some kind of conceptual representation might form for individual novel fonts in the present task. However, false recognition of pictures of objects resembling only single studied exemplars has also been demonstrated, supporting the suggestion that perceptual similarity itself can drive false recognition (*Stahl, Henze & Aust, 2017*, May 15).

Support for increased perceptually-driven false recognition in aging also comes from several studies where multiple studied words are related to phonologically similar lures (*Budson et al., 2003*; *Sommers & Huff, 2003*; *Watson, Balota & Sergent-Marshall, 2001*; see also *Rankin & Kausler, 1979*), although not in a recent study in which lures were only related to single studied words (*Ly, Murray & Yassa, 2013*). As for pictures, an age-related increase in false recognition of perceptually similar verbal lures may be present only when multiple related items have been studied. However, for phonological lures this is unlikely to be driven by formation of a novel conceptual representation, or by idiosyncratic categorical processing. The present data extend these findings by demonstrating greater false recognition of visually similar lures in older adults under conditions in which multiple items have been studied. They support the notion that older adults are more sensitive to recognition based on perceptual gist representations of sets of related but—unlike in the case of the phonologically similar lures—pre-experimentally unfamiliar features.

In contrast to the findings just discussed for the Font Only condition, we found age-invariant associative false recognition, both in Experiment 2 alone and across the two experiments. This invariance was supported by Bayes Factors for null effects of age in the Correlated Font condition. Critically, false recognition by older adults was not greater in this condition than in the Font Only condition, as originally predicted. Although

older adults often show increased false recognition in the DRM paradigm, a number of previous studies have also found no such effect. Across studies published prior to 2006 in which recognition was not measured after an initial recall test, *Gallo (2006)* found a non-significant (3%) mean increase in false recognition, although there was a significant (7%) mean increase in false recall. A key factor determining whether there are age effects is thought to be impairments in retrieval monitoring, demands for which are greater for recall than recognition and which may also be encouraged by list blocking (*Gallo, 2006*). A second modifying factor is thought to be the degree to which the experimental task encourages reliance on gist as opposed to specific memory (*Koutstaal, 2006*; *Tun et al., 1998*). *Tun et al. (1998)* found that older adults were more susceptible to false recognition only when study lists were blocked, unlike in the present study. Although our finding of age-invariant associative false recognition does not rule out age-related increases in blocked tasks, it is notable that we still found an age-related increase in false recognition of perceptually related lures.

In Experiment 2, the results for the Unique Font condition (in which lures differed in their font from all but one studied associate) were inconclusive. Overall and adjusted false recognition measures showed non-significant (but not clearly null) age effects which were intermediate between those for the Correlated Font (in which lures were similar in terms of their font to all studied associates) and Font Only conditions. This was not the clear pattern of age-invariant performance observed in the Correlated Font condition and therefore limits the generality of our conclusion that associative false recognition was not increased in older people in the present study. It will be important in future studies to compare age effects in a Correlated Font condition with those in a DRM condition with a standard simple font.

Another potential influence on both semantically- and perceptually-related false recognition in older adults is differential responding to the font information between age groups. One possibility is that older adults misunderstood the instructions and responded instead to identify test items in familiar fonts as studied. This strategy would have given rise to very high false alarm rates in older adults in the Correlated Font as well as the Font Only condition, but not in the Unique Font condition in Experiment 2. Under these conditions, we should have observed age-related increases in adjusted false recognition in the Correlated Font and Font only conditions relative to the Unique Font condition, a pattern not supported by the data. Extreme values of adjusted false recognition were also not markedly more common in the older group: in Experiment 1, 3 young and 4 older showed values of $>= .5$, and in Experiment 2, 7 young and 8 older showed FA values of $>= .75$ (see Tables 1 and 2). Three older individuals across the two studies did show high overall FA (to Novel as well as to studied-font conditions), but excluding them from analysis did not change the overall pattern of findings (see False recognition across experiments). Another possibility was group differences in the ability to avoid memory errors by systematically selectively attending to one dimension of the stimuli. In the present paradigm, because the perceptual similarity between words printed in the same fonts is task-irrelevant in the test phase, it would be beneficial to ignore the fonts as far as possible, and emphasize processing of the words' identity alone. Older adults may have been less able

to do this, consistent with evidence that they often fail to suppress irrelevant information (*Hasher, Quig & May, 1997*; *Campbell, Hasher & Thomas, 2010*; *Devitt & Schacter, 2016*). In the categorized pictures paradigm, on the other hand, it is advantageous to attend to the perceptual information. However, despite this difference, the present data together with those of *Pidgeon & Morcom (2014)* suggest that there is greater perceptually-related false recognition in older than young adults in both paradigms (see below).

If older adults are always more susceptible to perceptual similarity between lures and studied items, the age-related increase in false recognition in the Correlated Font condition should have been as large or larger than in the Font Only condition, at least if associative and perceptual effects are independent. Both age groups falsely recognized more Correlated Font lures than Unique Font lures, replicating *Arndt & Reder*'s (*2003*) results despite the intermixed presentation. However, shared font did not boost false recognition of critical lures specifically in older adults. One reason for this could be that older adults did not process font information as strongly when items were associatively related. Although encoding processing was presumably similar since conditions were intermixed, it is possible that in older adults, rapid high-confidence recognition of the critical lures at retrieval led to reduced incidental processing of font information. A second, perhaps more plausible, possibility derives from the finding that in young adults, false recognition of critical lures is specifically enhanced when they are presented in the same font as their studied associates, as opposed to when they are presented in the same font as other studied items (*Arndt, 2010*). This must reflect encoding of the association between font and study list, and there is strong evidence that associative memory is impaired in aging (*Naveh-Benjamin, 2000*). Thus, older adults' false recognition may have lacked this boost. Note that both these proposals contradict the above suggestion that young adults may have been better at ignoring font information at test.

Our starting point was a comparison with earlier work using the categorized pictures paradigm (*Koutstaal et al., 2003*; *Pidgeon & Morcom, 2014*). The present results suggest that the determinants of older adults' reliance on gist and/or prior knowledge may differ from those in the DRM paradigm. To our knowledge, all published studies have found substantial increases in older adults' false recognition of categorically-related lures, even though conditions are intermixed at encoding and retrieval (e.g., *Koutstaal et al., 1999a*; *Koutstaal et al., 1999b*; *Koutstaal, 2003*; *Koutstaal et al., 2003*; *Koutstaal & Schacter, 1997*; *Lovden, 2003*; *Pidgeon & Morcom, 2014*; *Rankin & Kausler, 1979*). Furthermore, older adults show greater increases in false recognition of lures from familiar 'concrete' categories than of unfamiliar 'abstract' categories (*Koutstaal et al., 2003*; *Pidgeon & Morcom, 2014*). It is therefore possible that prior semantic knowledge does specifically impact memory errors in older adults but does so predominantly via similarity, rather than via association. Older adults are also more likely to falsely recognize phonologically-similar lures than the young (although in those studies older groups also showed increased associative false recognition, unlike here; *Budson et al., 2003*; *Watson, Balota & Sergent-Marshall, 2001*). Another factor differing between DRM and categorized pictures paradigms may be the relative influences of the multiple types of semantic and associative relations between studied items and lures

which have been shown to contribute to false recognition (e.g., *Cann, McRae & Katz, 2011*; *Coane et al., 2015*; *McEvoy, Nelson & Komatsu, 1999*; *Montefinese, Zannino & Ambrosini, 2014*). Systematic evaluation of these variables in needed to understand the specific mediators of the sometimes very pronounced effects of age on false memory.

## Conclusions

The present study examined false recognition of associatively related and perceptually similar lures in older and younger adults. While older adults did not show an increase in false recognition of critical lures, they were more likely to falsely recognize lures which were printed in the same font as studied items. The results suggest that increases in false memory in aging are not always driven by greater reliance on prior knowledge, but that older adults show increased false recollection which can reflect increased susceptibility to perceptual resemblance between novel and previously encountered occurrences.

### Abbreviations

| | |
|---|---|
| **DRM** | Deese, Roediger & McDermott |
| **M** | mean |
| **SD** | standard deviation |
| **ANOVA** | analysis of variance |
| **MSE** | mean square effect |
| **BF** | Bayes factor |

## ACKNOWLEDGEMENTS

The authors are grateful to Rebecca Williams for assistance with data collection. Alexa M. Morcom is a member of the University of Edinburgh Centre for Cognitive Ageing and Cognitive Epidemiology (CCACE), part of the cross-council Lifelong Health and Wellbeing Initiative (MR/K026992/1).

### Funding

The authors received no funding for this work.

### Competing Interests

The authors declare there are no competing interests.

### Author Contributions

- Kayleigh Burnside and Caroline Hope performed the experiments, analyzed the data, contributed reagents/materials/analysis tools, wrote the paper, reviewed drafts of the paper.
- Emma Gill performed the experiments, contributed reagents/materials/analysis tools, wrote the paper, reviewed drafts of the paper.
- Alexa M. Morcom conceived and designed the experiments, analyzed the data, contributed reagents/materials/analysis tools, wrote the paper, prepared figures and/or tables.

## Human Ethics

The following information was supplied relating to ethical approvals (i.e., approving body and any reference numbers):

The study was approved by the Psychology Research Ethics Committee, University of Edinburgh.

## Data Availability

Burnside, Kayleigh; Hope, Caroline; Gill, Emma; Morcom, Alexa. (2017). Effects of perceptual similarity but not semantic association on false recognition in aging, 2013-2016 [dataset]. University of Edinburgh, School of Psychology, Philosophy and Language Sciences. http://dx.doi.org/10.7488/ds/2233.

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
