# Peer review of "Effects of perceptual similarity but not semantic association on false recognition in aging"

_PeerJ, doi:10.7717/peerj.4184_

## Round 0.1 · original submission · Major Revisions

The reviewers' recommendations were different although their comments overlap. Both reviewers had some difficulty understanding the complexities of the experiments and this reduced their enthusiasm for your article, particularly in the case of Reviewer 1 who was more critical.

In your revision, please pay close attention to the ability of readers to follow your approach even if they are not familiar with the experimental design that you adopt from past work. It might help if you add a design overview at the start of Experiment 1, which explains the conditions. I agree with Reviewer 1 that a figure illustrating the conditions would help. I also think that explaining the take-home message from each effect or manipulation in the discussions for each experiment would help readers to see the significance of your results. At the moment the results are succinctly summarised in terms of conditions, but the relevance of these conditions to your argument isn't re-stated.

The reviewers also have some concerns about your statistical analysis. Reviewer 2's comments about BF reporting seem particularly pertinent.

Reviewer 1 ·

Basic reporting

The crucial summary of the design in Experiment 1 appears to have a grammatical error (line 100); also what is a list lure? If it was in the list is it a lure or a target? We only find out at line 109. The method may benefit from a design section that is self-contained. I found the method very hard to follow and I feel that it would benefit from figures depicting each of the three conditions.

I am unfamiliar with the way that interactions are reported intertwined with main effects and cannot discriminate between original analyses, follow-up analyses and analyses based on different dependant variables. Nested parentheses caused further confusion.

The y-axis is not labelled on the Figure. All four panels on the Figure have different maximum y-values, which hinders comparison.

Experimental design

I could not fully follow how words were placed into each condition and into study and test list in the 2nd paragraph of the material section in Experiment 1. Is the word ‘novel‘ being used to label non-studied words on line 113 or does this refer to the fonts? There seems to be sufficient detail to replicate if one was prepared to devote a significant amount of time to unpack the information.

I did not see any control for the degree of congruency between fonts and words, which could have differed between study and test items and between conditions.

Validity of the findings

My lack of comprehension of what was actually being measured restricts my comments on the validity of the statistics although individual F-values appear to have been reported correctly. The Bayes factor analyses were a good way to account for multiple tests. The general discussion is fairly long and is therefore not always closely linked to the results.

Additional comments

In some respects I feel that I may have missed parts of the article due to my lack of understanding, particularly of the results sections. However, I likely made more effort with the paper as a reviewer than would a normal reader and the lack of clarity in the article prohibits my recommendation for publication.

It seems as though the main point of interest is age differences in perceptual (font) versus semantic (associative strength) lures. However, Experiment 1 had the same perceptual presentation across all 3 manipulations of semantic relatedness. Experiment 2 controlled for this by also manipulating font uniqueness and the take home message is that perceptual lures can lead to greater age deficits than semantic lures. As far as I understand it, this could be conveyed in a much briefer write-up that excludes a full statistical exploration of the data and that has a shorter discussion.

Reviewer 2 ·

Basic reporting

The paper is generally very well-written and structured.

Minor issues:
- The authors could expand on the relevance of remember/know judgments in the introduction (l.64).
- Figure 1, together with its caption, should be more self-explanatory; please add Y-axis labels.
- l. 173-174 missing p-values
- l. 183 incorrect p-value
- l. 315 should be table 2
- Table 2: How was false recognition to font only lures in older adults computed/corrected? Shouldn't this value be closer to .38?

Experimental design

The study seems carefully designed. Yet, for the author to fully understand the design the clarity of reporting the methods of Experiments 1 and 2 could be improved (e.g., how many lists and items of each type were used and how exactly were they combined to create the materials for each condition; perhaps a table could more easily illustrate the construction of materials; an appendix could be addedd that provides the full list of words and fonts to increase replicability).

Validity of the findings

My major concern is that, instead of false memory, the results might reflect strategic differences between age groups in how they (mis)understood the recognition instructions. There was no instructions regarding the role of fonts at test. Did age groups differ in their understanding of the relevance of fonts to recognition judgments? I.e., older adults may have understood the similarity between studied and test fonts as relevant (instead of to-be-ignored) for the recognition test. If so, their recognition of lures would no longer be interpretable as "false" memory.

A perhaps related puzzling finding is that, in Experiment 2, older adults had increased perceptual false memory (font only condition), as well as increased semantic false memory (unique font condition) - but when both factors were combined (correlated font condition) the age-related false memory increase was eliminated. This needs to be discussed, perhaps with the differences in true recognition in mind.

A third issue concerns statistical analyses. To interpret null results, the authors report Bayes Factors in some cases, which is helpful for interpretation. In other cases, however, they still interpret null findings without reporting BF, especially for ANOVA analyses. I would recommend to add the missing BFs that are easily computed from JASP (or to avoid interpreting the respective null findings).
In interpreting BF values, the authors unfortunately decide that a BF > 3 represents strong evidence, but that a BF < 3 only yields inconclusive evidence. I believe there are two problems with this interpretation: first, there is little agreement that BF > 3 is strong (e.g., http://www.nicebread.de/grades-of-evidence-a-cheat-sheet/). Second, the BF value is a continuous measure of evidence (or relative belief updating), and lends even less support for introducing arbitrary cutoffs or decision criteria than the p value; we all know that the difference btw significant and non-significant is itself not significant, and analogously (but for different reasons) a small difference in BF (e.g., between 3.3 and 2.1) is itself not (strong) evidence for a difference in support for two hypotheses - but such a difference might be directly tested by targeted model comparisons in some cases.
Finally, the authors report that they used robustness checks to test whether their result patterns depend on the choice of priors. They do not, however, report detail on these analyses or their results.

---

## Round 0.2 · Minor Revisions

The reviewers agree that this version is clearer and will be of interest to researchers in the field. Reviewer 2 has requested some further important revisions, particularly to the results and interpretation. I hope you will be able to make these changes. If I can see these changes are thorough, I hope to be able to make a final decision without the need for re-review.

Reviewer 1 ·

Basic reporting

Improved (see below)

Experimental design

Same as original submission.

Validity of the findings

Same as original submission.

Additional comments

I feel that the clarity of the article has improved and that there is a clear take-home message that is sufficiently evidenced and of interest to ageing researchers.

The new figure aided clarity. I still feel that this simple message could be presented with fewer statistics and measures but I was better able to follow this version.

Reviewer 2 ·

Basic reporting

The authors have added Bayes factors for some non-significant findings that are interpreted but not for all of them. Please supply additional analysis for...

- Both experiments:
- non-significant difference in forward digit span between age groups
- False recollection of novel items

- Experiment 1:
- adjusted false recognition report BF for non-significant post hoc tests

- Experiment 2:
- True recognition: report BF for absence of interaction effect
- True recollection: Report statistics for group and interaction effects including BF for non-significant findings.
- Also, double check that the reported BF = 0.96 is correct (seems surprising given the strong effect in the frequentist analysis, partial eta² = .25, p < .001)

- Experiment 2, adjusted false recognition: Double check BF for effect of interaction given that the frequentist analysis was non-significant.

Experimental design

no further comments

Validity of the findings

- The authors have clarified in the method section that participants received no instructions regarding the relevance of font. They rightly state that it is unlikely that older participants responded solely based on fonts. However, as previously pointed out the possibility remains that older adults may have considered the font relevant in addition to the words and more so than younger participants. In fact, the possibly impaired ability to ignore fonts that the authors discuss in lines 528-539 makes a similar point. Strategic or involuntary processing of fonts both have consequences for the interpretation of the findings. In case of strategic differences it would be hard to attribute the findings to "false" memory. This caveat to the interpretation should be discussed.

- The authors interpret the age-related increase in perceptual false recognition as being caused specifically by false recollection---being accompanied by a recollective experience rather than feelings of familiarity due to semantic knowledge---(e.g. l. 219-221 or 457-459) because the analysed "remember" responses follow the false recognition pattern. To validly make claim that recollection is selectively driving the effects, the authors would need to show that the same pattern is absent in know responses. Moreover, in the analysis of false recollection in Experiment 1 the ANOVA is followed up by post hoc tests of age effects for each condition. However, it appears that, contrary to all other analyses, the authors do not adjust the alpha level for multiple comparisons (alpha = .017). Had they done so, the age-effect in the font only condition would not reach statistical significance. This would further weaken the claim that the effect is driven by recollection.

- After correct adjustment of false recognition rates in Experiment 2, the group x condition interaction is no longer significant and the evidence for the interaction effect is anecdotal. Hence, the interpretation of differences between subsequently significant and non-significant post hoc tests of age effects per condition is questionable. Similarly, in the subsequent across-experiment analysis the authors report extreme evidence for the group x condition interaction but do not report whether they found evidence against a three-way interaction with experiment. Given the ambiguous results of Experiment 2 such an interaction would call into question whether the group x condition interaction was indeed replicated.

Additional comments

- We appreciate the authors' sharing of raw data and stimulus materials.
- The authors may consider moving lines 144-167 to a data-analysis subsection of the methods section.

---

## Round 0.3 · accepted · Accept

I have reviewed the changes you have made in response to the additional points raised by Reviewer 2. I am happy with the way these have been addressed and therefore don't see a need for re-review. Thanks for submitting your interesting paper to PeerJ.